# Differentiating the Effects of Streamflow and Topographic Changes on the Water Level of Dongting Lake, China, Using the LSTM Network and Scenario Analysis

**Jihu Zou** [1], **Feng Huang** [1,*], **Feier Yu** [1], **Xingzhi Shen** [2], **Shuai Han** [2], **Zhan Qian** [2] and **Heng Jiang** [2]

1   College of Hydrology and Water Resources, Hohai University, Nanjing 210098, China; zoujihu1999@163.com (J.Z.); starshine268@163.com (F.Y.)
2   Hunan Water Resources and Hydropower Survey, Design, Planning and Research Co., Ltd., Changsha 410007, China; xingzhishen2023@126.com (X.S.); hsh1109@163.com (S.H.); zhanqian0920@126.com (Z.Q.); jiangheng01@126.com (H.J.)
*   Correspondence: huangfeng1987@hhu.edu.cn; Tel.: +86-025-8378-6621

**Abstract:** Dongting Lake is the second largest freshwater lake in China and an internationally critical habitat for migratory birds. However, after 2004, the multiyear mean water levels of West Dongting Lake (WDL), South Dongting Lake (SDL), and East Dongting Lake (EDL) in the high-water stage decreased by 1.05 m, 1.15 m, and 1.32 m, respectively. Different areas of Dongting Lake experienced various degrees of shrinkage. It is necessary to study the dominant driving factors and their contributions to the falling water level. In this study, the water level changes in Dongting Lake were analyzed, and a long short-term memory neural network model was constructed to simulate the water level of Dongting Lake. Moreover, the contribution of changes in streamflow and topographic conditions to the water level changes in different areas of Dongting Lake was estimated with scenario analysis. The research results show that the changes in the streamflow were the main driving factors for the water level decline of WDL, SDL, and EDL in the high-water stage, and their contributions were 0.74 m, 0.97 m, and 1.16 m, respectively. The topographic changes had a great falling effect on the water level of Dongting Lake, and the falling effect on the water levels from October to June of the following year was the strongest in EDL (0.81 m), followed by WDL (0.49 m), and the weakest in SDL (0.3 m). These results can provide a scientific reference for the management of the water resources of Dongting Lake.

**Keywords:** water level changes; contribution; changes in streamflow; changes in topographic conditions; long short-term memory neural network; Dongting Lake

## 1. Introduction

Dongting Lake, located in the middle reaches of the Yangtze River, is the second largest freshwater lake in China. In recent decades, under the influence of climate change and human activities, the water level of Dongting Lake has changed significantly, which has had a significant impact on the health of the lake's ecosystem and the people's lives in the surrounding areas [1]. Studies have shown that, after the operation of the Three Gorges Reservoir (TGR) began, the streamflow of Sankou (Songzikou, Taipingkou, and Ouchikou), which flows from the Yangtze River into Dongting Lake, significantly reduced [2]. After 2003, the water level of Dongting Lake decreased, especially in the extremely dry year of 2011, when the water surface area of Dongting Lake was greatly reduced, the dry season was two months longer than before, and more than 1000 hectares of wetlands dried up [3,4]. A reduction in the water level of Dongting Lake will lead to a reduction in wetland area, an ecosystem degradation, and the habitat loss of migratory birds, and, at the same time, the growth and reproduction of birds, fish, and macroinvertebrates will be greatly affected



by the reduced water level [5–9]. Hence, it is essential to determine the factors that cause water level reductions in Dongting Lake and their level of contribution.

The Dongting Lake receives inflows from Sishui (Xiangjiang River, Zishui River, Yuanjiang River, and Lishui River) of Hunan Province and Sankou, and it connects with the Yangtze River at the outlet of the lake. The factors that affect the water level of Dongting Lake can be summarized in the following aspects. (1) The operation of the TGR: The TGR intercepts floods during the flood season and discharges water to generate electricity during the dry season, which causes the water level of the downstream river to change and further changes the streamflow of Sankou into Dongting Lake, thus affecting the water level of Dongting Lake [10,11]. In addition, the TGR intercepts a large amount of sediment, resulting in a decrease in sediment concentration in downstream rivers. With changes in the amount of sediment entering Dongting Lake from the Yangtze River, variations in erosion and deposition will inevitably occur in Dongting Lake. (2) Changes in the streamflow of Sishui: Sishui is one of the inflows of Dongting Lake. An increase and decrease in its streamflow will directly lead to the rise and fall of the water level of Dongting Lake. (3) Changes in climatic conditions: for example, changes in precipitation conditions will affect the streamflows of Sankou and Sishui, thereby affecting the water level of the Dongting Lake [12]. (4) Water intake and utilization in the lake area: with the development of social economy [13], the water level of Dongting Lake will be affected by the water intake and utilization in the lake area due to human production and living activities. (5) Human activities: These include the reclamation of lakes and sand mining. This will cause changes in the shape of rivers and lakes, affect the water storage capacity of the lakes, and therefore cause changes in the water level [14]. For the Dongting Lake Basin, changes in precipitation conditions mainly affect the runoff, thereby affecting the water level of Dongting Lake. The main factors affecting the water level of Dongting Lake can be divided into three categories: the changes in streamflow, the changes in topographic conditions, and other factors. This study focuses on the effects of changes in streamflow and topographic conditions. The changes in streamflow here are caused by the streamflow changes in Sishui and the Yangtze River. The changes in topographic conditions here are mainly caused by sand mining activities and the change in erosion and deposition in Dongting Lake. However, the mechanism of these effects is still unclear.

Huang et al. [15] used a back propagation (BP) neural network to analyze the impacts of streamflow and topographic changes on the water level of Poyang Lake in the dry season, but the many-to-one neural network structure is not applicable to Dongting Lake, and the effects of streamflow and topographic changes on the water level of WDL, SDL, and EDL in different seasons are completely different. Dongting Lake has a more complex water system than Poyang Lake. The water of the Yangtze River flows from Sankou to different areas of Dongting Lake, Sishui also flows into different areas of Dongting Lake, and the water finally flows into the Yangtze River at Chenglingji. The water levels of WDL, SDL, and EDL are different in each season, so they cannot be analyzed as a whole. The relationship between the changes in streamflow and topographic conditions and the changes in water level needs to be clarified.

This study mainly explores the contribution of different driving factors to the water level changes in the different areas of Dongting Lake and takes into account the important influence of the changes in topographic conditions. The changes in the water level of Dongting Lake were analyzed based on long-term observed hydrological data. By simulating the water level of Dongting Lake using a long short-term memory neural network, the contribution of changes in streamflow and topographic conditions to the water level changes in different areas of Dongting Lake was analyzed. This study can provide a reference for the analysis of the causes of water level changes in Dongting Lake.

## 2. Study Area and Data

### 2.1. Study Area

The Dongting Lake Basin (Figure 1) covers an area of about 260,000 km², accounting for about 14% of the area of the Yangtze River Basin. Dongting Lake is mainly composed of WDL, SDL, and EDL. The large capacity for water storage of Dongting Lake is important for the regulation of flooding in the Yangtze River Basin [16]. The Dongting Lake Basin is located in a subtropical monsoon climate zone, with an average annual rainfall of about 1400 mm [17]. The water level of Dongting Lake varies significantly with the seasons. April to June is the water-rising period, July to September is the high-water stage, October to November is the water-falling stage, and the dry season lasts from December to March of the following year [5,18].

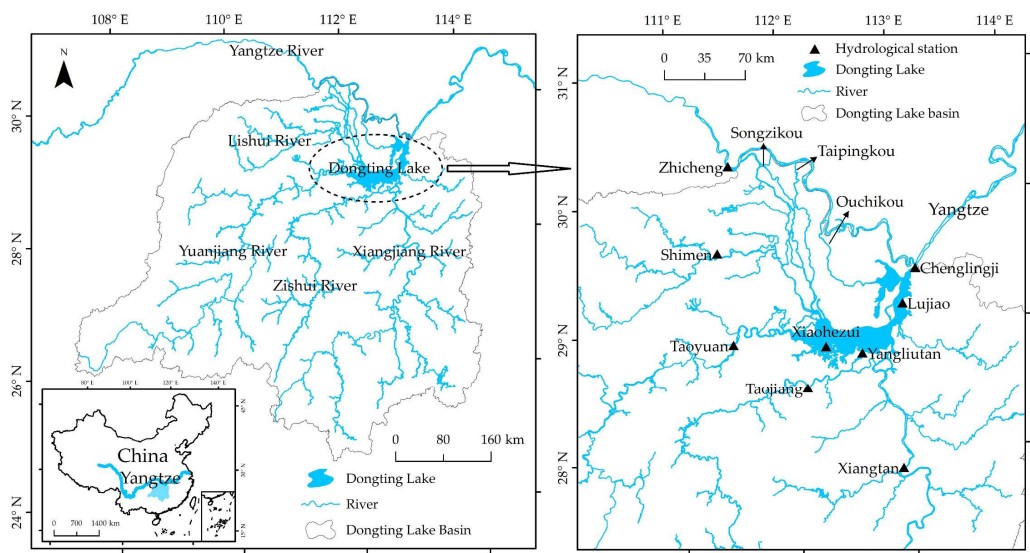

**Figure 1.** Distribution of observation stations in the Dongting Lake Basin.

### 2.2. Hydrological Data

Fourteen hydrological stations (Table 1) were selected as data-gathering points for this study. Three hydrological stations (Xiaohezui in WDL, Yangliutan in SDL, and Lujiao in EDL) monitor the water level of Dongting Lake. One hydrological station (Chenglingji in EDL) monitors the outflow of Dongting Lake. Four hydrological stations (Xiangtan, Taojiang, Taoyuan, and Shimen) monitor the streamflow of Sishui. Five hydrological stations (Xinjiangkou, Shadaoguan, Mituosi, Kangjiagang, and Guanjiapu) monitor the streamflow of Sankou. One hydrological station (Zhicheng) monitors the streamflow of the Yangtze River. The observed water level (m) and streamflow (m³/s) data were obtained from the China Yangtze River Water Resources Commission, who conducted a rigorous inspection of the recorded data to ensure its availability. The period of daily observed hydrological data was from 1992 to 2019.

**Table 1.** Hydrological stations in the Dongting Lake Basin.

| Hydrological Station | Location | Period | Data Type |
|---|---|---|---|
| Xiaohezui | West Dongting Lake | 1992–2019 | Water level |
| Yangliutan | South Dongting Lake | 1992–2019 | Water level |
| Lujiao | East Dongting Lake | 1992–2019 | Water level |
| Chenglingji | East Dongting Lake | 1992–2019 | Streamflow |
| Xiangtan | Xiangjiang River | 1992–2019 | Streamflow |
| Taojiang | Zishui River | 1992–2019 | Streamflow |
| Taoyuan | Yuanjiang River | 1992–2019 | Streamflow |
| Shimen | Lishu River | 1992–2019 | Streamflow |

**Table 1.** *Cont.*

| Hydrological Station | Location | Period | Data Type |
| --- | --- | --- | --- |
| Xinjiangkou | Songzikou | 1992–2019 | Streamflow |
| Shadaoguan | Songzikou | 1992–2019 | Streamflow |
| Mituosi | Taipingkou | 1992–2019 | Streamflow |
| Kangjiagang | Ouchikou | 1992–2019 | Streamflow |
| Guanjiapu | Ouchikou | 1992–2019 | Streamflow |
| Zhicheng | Yangtze River | 1992–2019 | Streamflow |

## 3. Methods

### 3.1. Detection of Abrupt Change in Water Level Using Heuristic Segmentation Algorithm

The heuristic segmentation algorithm is an abrupt change detection method using mean values proposed by Bernaola-Galván et al. [19]. This method can effectively divide the original sequence into multiple sub-periods, whose mean values are significantly different. The detection method can overcome the requirements of traditional statistical testing methods for detecting sequence stationarity and linearity. This method is widely used in the study of hydrological processes [20,21]. This heuristic segmentation algorithm was applied to identify abrupt change points in the annual mean water level sequence of Dongting Lake.

### 3.2. Long Short-Term Memory Neural Network Techniques

Artificial neural networks (ANNs) can simulate complex hydrological systems and have been widely used in the field of hydrology [22]. Long short-term memory neural network (LSTMNN) is an improved recurrent neural network (RNN) with strong learning ability, which can capture the memory characteristics of time series and has a strong potential in processing complex and massive data [23]. Compared with RNN, LSTMNN can avoid the problem of gradient degradation and can achieve accurate modeling of time series with long-term dependencies. This method consists of an input layer, hidden layer, and output layer and can be used to solve the input–output mapping problem. The structural units of LSTMNN include the input gate, the forget gate, and the output gate. When the information passes through these three gate structures, it can be selectively processed to realize the storage and updating of the information [24,25]. Neural network technology has been widely used in many fields [26], and it is an effective method to simulate the long-term change process of the Dongting Lake water level [3,27].

### 3.3. Water Level Simulation Using LSTMNN

For the simulation of the daily water level of Dongting Lake, a three-layer network was applied, including an input layer, a hidden layer, and an output layer. The input layer contained the streamflow of the Yangtze River monitored at Zhicheng station; the streamflows of Sishui monitored at Xiangtan station, Taojiang station, Taoyuan station, and Shimen station; and the previous day's water levels of Dongting Lake at Xiaohezui station, Yangliutan station, and Lujiao station. The output layer contained the day's water levels at Xiaohezui station, Yangliutan station, and Lujiao station (Figure 2). The specific process of the model when simulating the water level is as follows: The water levels of Xiaohezui station, Yangliutan station, and Lujiao station on the first day and the streamflows of Zhicheng station, Xiangtan station, Taojiang station, Taoyuan station, and Shimen station on the second day enter the input layer, and after processing with the structural unit of LSTMNN, the simulated water levels of Xiaohezui station, Yangliutan station, and Lujiao station on the second day are obtained. The water levels of Xiaohezui station, Yangliutan station, and Lujiao station on the second day and the streamflows of Zhicheng station, Xiangtan station, Taojiang station, Taoyuan station, and Shimen station on the third day enter the input layer, and after processing with the structural unit of LSTMNN, the simulated water levels of Xiaohezui station, Yangliutan station, and Lujiao station on the third day are obtained, etc. The daily simulated water levels of Xiaohezui

station, Yangliutan station, and Lujiao station were finally obtained. The performance of LSTMNN was evaluated using Pearson Correlation Coefficient (R) and Root Mean Square Error (RMSE).

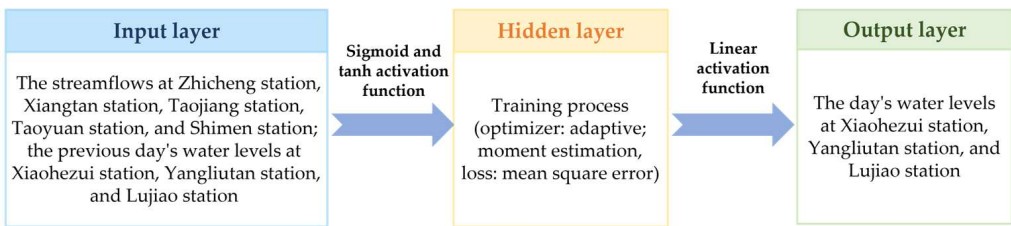

**Figure 2.** The structure of LSTMNN for simulating water level.

Taking the abrupt change point of the annual mean water level sequence of Dongting Lake as the segmentation point, the period from 1992 to 2019 was divided into two sub-periods. The sub-period before the abrupt change was named the pre-TGR period, and the sub-period after the abrupt change was named the post-TGR period. In addition, to ensure that comparisons and analyses can be performed in subsequent studies, the pre-TGR period and post-TGR period were adjusted appropriately so that they contained the same amount of data. An LSTMNN model was constructed using the daily hydrological data from the pre-TGR period and named the pre-TGR model. Then, another LSTMNN model with the same structure was constructed using the daily hydrological data from the post-TGR period and named the post-TGR model. These two models reflect the responses of the Dongting Lake water level to the streamflows of Sishui and the Yangtze River in two periods. For specific input conditions, the difference in the responses of the pre-TGR model and the post-TGR model can be considered to be caused by changes in topographic conditions. For each period of long-term hydrological data, the dataset was divided into training (the previous 75% data) and testing (the latter 25% data) datasets.

The trained pre-TGR model was further tested using daily hydrological data from the pre-TGR period. In addition, the trained post-TGR model was further tested using daily hydrological data from the post-TGR period. In this test, the simulation of the Dongting Lake water level used the simulated water level of the previous day as the input, but the initial simulation used the measured water level of the previous day as the input.

*3.4. Scenario Analysis to Calculate the Contribution of Changes in Streamflow and Topographic Conditions*

For Xiaohezui station, Yangliutan station, and Lujiao station, the water level changes can be expressed as follows:

$$\Delta H = H_{\text{post-TGR}} - H_{\text{pre-TGR}} \tag{1}$$

where $\Delta H$ represents the water level changes during the pre-TGR and post-TGR periods; $H_{\text{post-TGR}}$ is the water level during the post-TGR period; and $H_{\text{pre-TGR}}$ is the water level during the pre-TGR period.

Under the influence of streamflow and topographic conditions, the changes in the water level can be estimated using the following method:

$$\Delta H = \Delta H_{\text{streamflow}} + \Delta H_{\text{topography}} \tag{2}$$

where $\Delta H_{\text{streamflow}}$ represents the changes in water level caused by the changes in stream-flow, and $\Delta H_{\text{topography}}$ represents the changes in water level caused by the changes in topographic conditions.

$\Delta H_{\text{streamflow}}$ can be estimated using the following method:

$$\Delta H_{\text{streamflow}} = \Delta H_{\text{Sishui}} + \Delta H_{\text{Yangtze River}} \tag{3}$$

where $\Delta H_{\text{Sishui}}$ represents the changes in water level caused by the changes in the streamflow of Sishui, and $\Delta H_{\text{Yangtze River}}$ represents the changes in water level caused by the changes in the streamflow of the Yangtze River.

The following four scenarios were designed to estimate the contribution of changes in streamflow and topographic conditions to water level changes in WDL, SDL, and EDL:

1. The pre-TGR period streamflows of Zhicheng station, Xiangtan station, Taojiang station, Taoyuan station, and Shimen station were input into the pre-TGR model, resulting in the simulated water levels ($H_{\text{pre-TGR}}$) of WDL, SDL, and EDL.
2. The pre-TGR period streamflow of Zhicheng station and the post-TGR period streamflows of Xiangtan station, Taojiang station, Taoyuan station, and Shimen station were input into the pre-TGR model, resulting in the simulated water levels ($H'_{\text{pre-TGR}}$) of WDL, SDL, and EDL.
3. The post-TGR period streamflows of Zhicheng station, Xiangtan station, Taojiang station, Taoyuan station, and Shimen station were input into the pre-TGR model, resulting in the simulated water levels ($H''_{\text{pre-TGR}}$) of WDL, SDL, and EDL.
4. The post-TGR period streamflows of Zhicheng station, Xiangtan station, Taojiang station, Taoyuan station, and Shimen station were input into the post-TGR model, resulting in the simulated water levels ($H_{\text{post-TGR}}$) of WDL, SDL, and EDL.

In Scenarios 1–4, the water level simulation of WDL, SDL, and EDL used the simulated water level of the previous day as the input, but the initial simulation used the measured water level of the previous day as the input.

By comparing and analyzing the four groups of simulated water levels in Scenarios 1–4, the contribution of different influencing factors to the changes in water level in WDL, SDL, and EDL can be evaluated. The differences between the output of Scenario 2 and Scenario 1 indicate the effect of the changes in the streamflow of Sishui; the differences between the output of Scenario 3 and Scenario 2 indicate the effect of the changes in the Yangtze River streamflow; the differences between the output of Scenario 3 and Scenario 1 indicate the effect of the changes in streamflow; the differences between the output of Scenario 4 and Scenario 3 represent the effect of the changes in topographic conditions. The changes in water level can be further expressed as follows:

$$\Delta H_{\text{Sishui}} = H'_{\text{pre-TGR}} - H_{\text{pre-TGR}} \tag{4}$$

$$\Delta H_{\text{Yangtze River}} = H''_{\text{pre-TGR}} - H'_{\text{pre-TGR}} \tag{5}$$

$$\Delta H_{\text{streamflow}} = H''_{\text{pre-TGR}} - H_{\text{pre-TGR}} \tag{6}$$

$$\Delta H_{\text{topography}} = H_{\text{post-TGR}} - H''_{\text{pre-TGR}} \tag{7}$$

## 4. Results

### 4.1. Detection of Abrupt Change in Water Level in Dongting Lake

The heuristic segmentation algorithm was used to analyze the abrupt change points of the annual mean water level sequence of Dongting Lake from 1992 to 2019. The results show that the abrupt change points of Xiaohezui station in WDL, Yangliutan station in SDL, and Lujiao station in EDL all occurred in 2004 (Figure 3), and the multiyear mean water levels of WDL, SDL, and EDL decreased by 0.49 m, 0.47 m, and 0.65 m, respectively, after 2004, which may have been greatly influenced by the operation of the TGR. Therefore, taking 2004 as the segmentation point, the research period was divided into two sub-periods: 1992–2003 and 2004–2019.

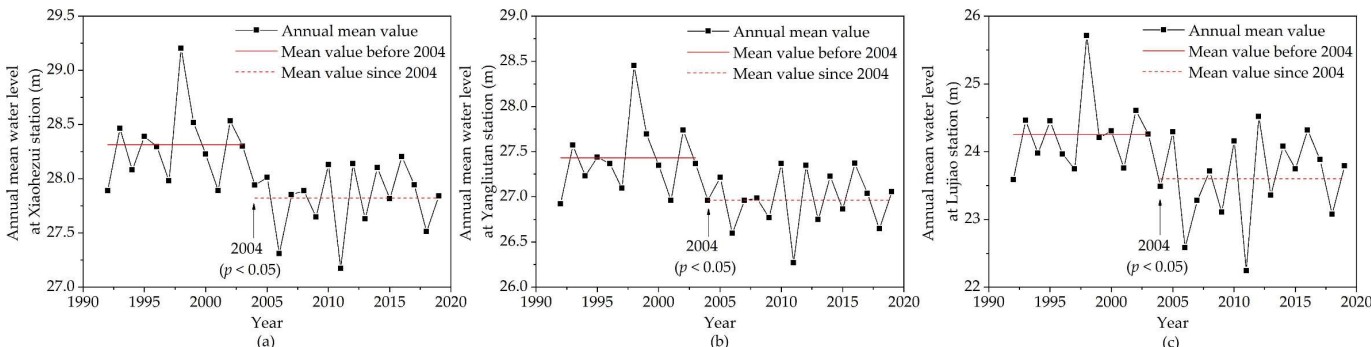

**Figure 3.** Detection of abrupt change in water level in Dongting Lake: (**a**) Xiaohezui station; (**b**) Yangliutan station; and (**c**) Lujiao station.

### 4.2. Training, Testing, and Verification of LSTMNN

To ensure that the two model input layers contained the same amount of data, the pre-TGR model was constructed using the daily hydrological data from 1992–2003 (the pre-TGR period), and the post-TGR model was constructed using the daily hydrological data from 2008–2019 (the post-TGR period). Figures 4 and 5 show the training and testing results of the pre-TGR and post-TGR models, respectively. In Figures 4 and 5, the simulated water level matches well with the observed water level, where the value of R is no less than 0.99, and the value of RMSE is no more than 0.16 m, indicating the good performance of the pre-TGR model and post-TGR model in simulating the water level of Dongting Lake.

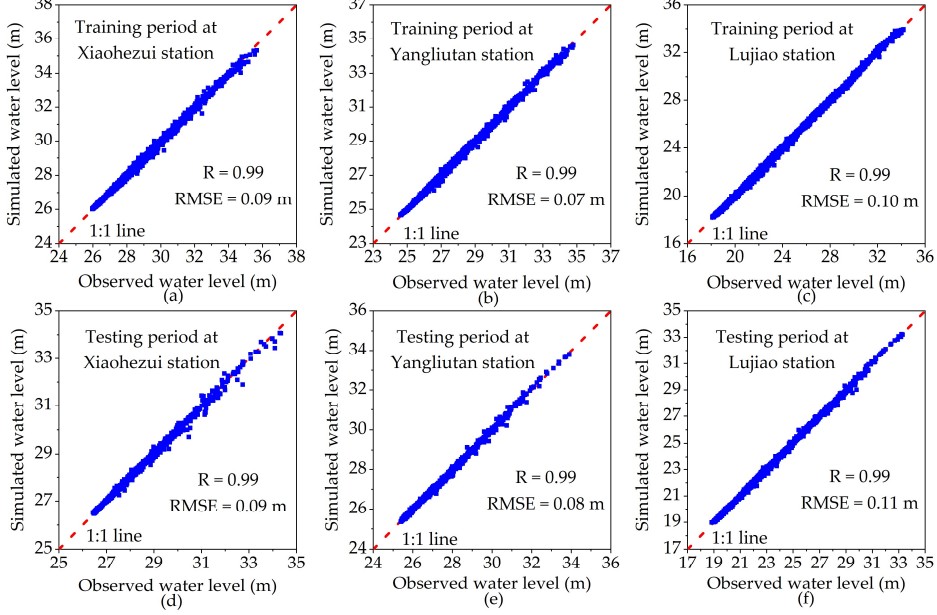

**Figure 4.** Scatter plots showing performances of training and testing of the pre-TGR model: (**a**) training period at Xiaohezui station; (**b**) training period at Yangliutan station; (**c**) training period at Lujiao station; (**d**) testing period at Xiaohezui station; (**e**) testing period at Yangliutan station; and (**f**) testing period at Lujiao station.

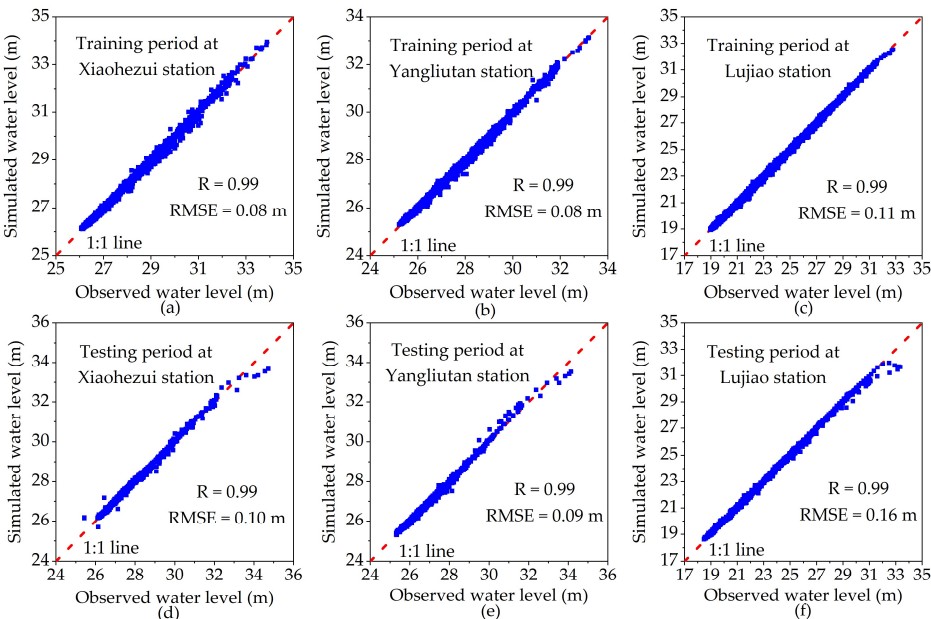

**Figure 5.** Scatter plots showing performances of training and testing of the post-TGR model: (**a**) training period at Xiaohezui station; (**b**) training period at Yangliutan station; (**c**) training period at Lujiao station; (**d**) testing period at Xiaohezui station; (**e**) testing period at Yangliutan station; and (**f**) testing period at Lujiao station.

The trained pre-TGR model was further tested using the daily hydrological data from the pre-TGR period (Figure 6). In addition, the trained post-TGR model was further tested using the daily hydrological data from the post-TGR period (Figure 7). In Figures 6 and 7, the value of R is no less than 0.99, and the value of RMSE is no more than 0.52 m. This shows the feasibility of using the LSTMNN model to analyze the contribution of different driving factors to the water level changes in Dongting Lake.

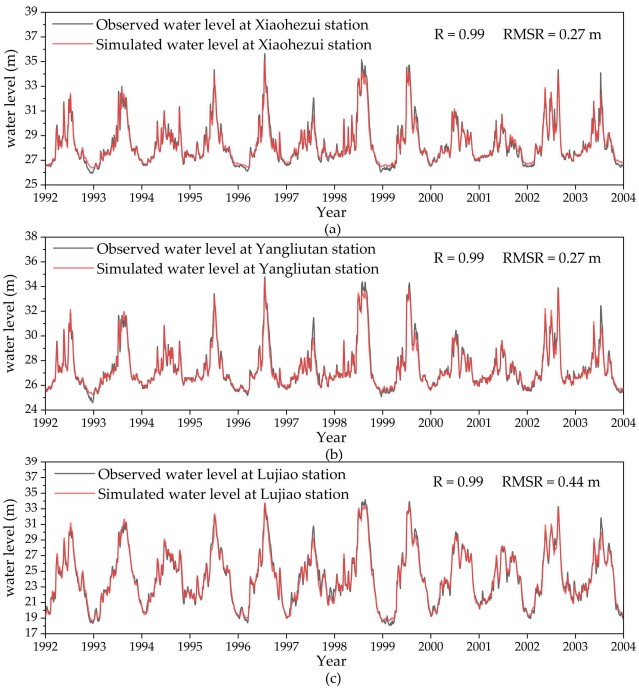

**Figure 6.** Simulation results of the Dongting Lake water level of the pre-TGR model: (**a**) Xiaohezui station; (**b**) Yangliutan station; and (**c**) Lujiao station.

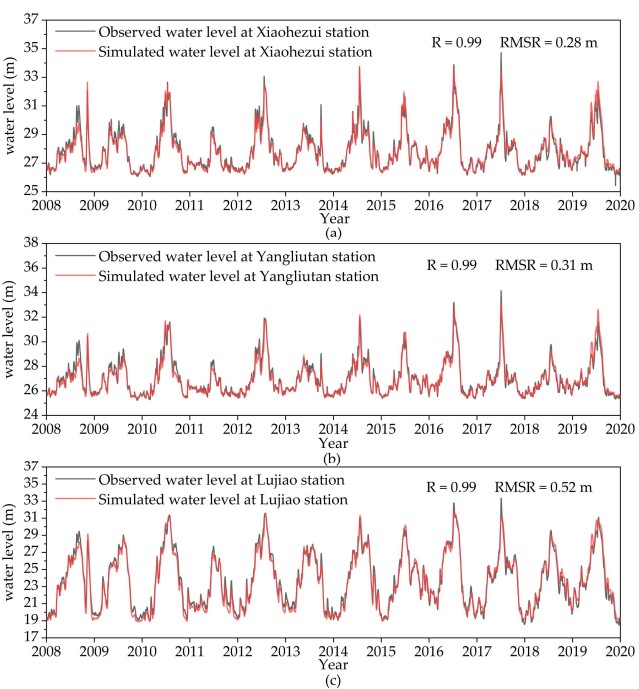

**Figure 7.** Simulation results of the Dongting Lake water level of the post-TGR model: (**a**) Xiaohezui station; (**b**) Yangliutan station; and (**c**) Lujiao station.

### 4.3. Hydrological Changes in the Dongting Lake Basin

Figure 8 shows the observed hydrological changes in Dongting Lake between the pre-TGR period and the post-TGR period, and the multiyear mean water levels of the pre-TGR period and the post-TGR period are the average values of 1992–2003 and 2008–2019. The comparison results show that the multiyear mean water levels at Xiaohezui station, Yangliutan station, and Lujiao station from July to September decreased by 1.05 m, 1.15 m, and 1.32 m, respectively. The multiyear mean water levels at Xiaohezui station, Yangliutan station, and Lujiao station from December to March of the following year decreased by 0.2 m, 0.17 m, and 0.2 m, respectively. The different areas of Dongting Lake experienced various degrees of shrinkage after 2004, and the shrinkage of EDL was the most serious.

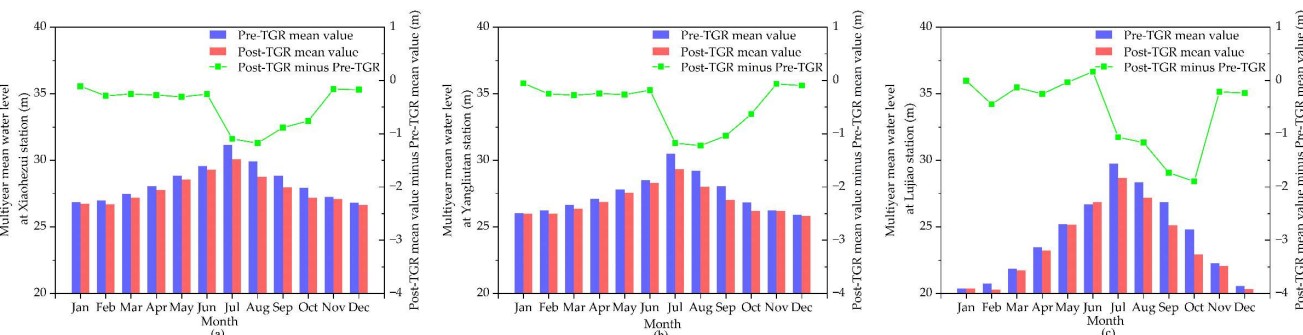

**Figure 8.** The observed multiyear mean water levels of Dongting Lake: (**a**) Xiaohezui station; (**b**) Yangliutan station; and (**c**) Lujiao station.

The streamflows of the Yangtze River, Sishui, and Sankou and the outflow of Dongting Lake are important factors affecting the water level of Dongting Lake (Figure 9). In the high-water stage, the multiyear mean streamflows of the Yangtze River and Sishui decreased by 3768 m³/s and 1548 m³/s, respectively, and this is closely related to the operation of the TGR and reservoirs upstream of Sishui, which store water during the high-water stage. The multiyear mean streamflow of Sankou and the outflow of Dongting Lake decreased



by 1563 m³/s and 3979 m³/s, respectively, in the high-water stage, which indicates that the streamflow of the Yangtze River into Dongting Lake and the outflow of Dongting Lake decreased during this period. The interaction between the Yangtze River and Dongting Lake changed. The multiyear mean streamflow of the Yangtze River increased by 2082 m³/s in the dry season, which shows that the release of water from the TGR to generate electricity during this period increased the downstream streamflow. Around May, the TGR releases water to free up storage for flood control, which also increases the downstream streamflow. In the dry season, the multiyear mean streamflow of Sishui decreased by 52 m³/s, the multiyear mean streamflow of Sankou increased by 62 m³/s, and the multiyear mean streamflow of the outflow of Dongting Lake decreased by 77 m³/s. The inflow of Dongting Lake is the sum of the streamflows of Sishui and Sankou. During the dry season, the inflow of Dongting Lake increased by 10 m³/s, and the outflow decreased by 77 m³/s, but the water levels of EDL, SDL, and EDL all decreased (Figure 8), which could be explained by topographic changes.

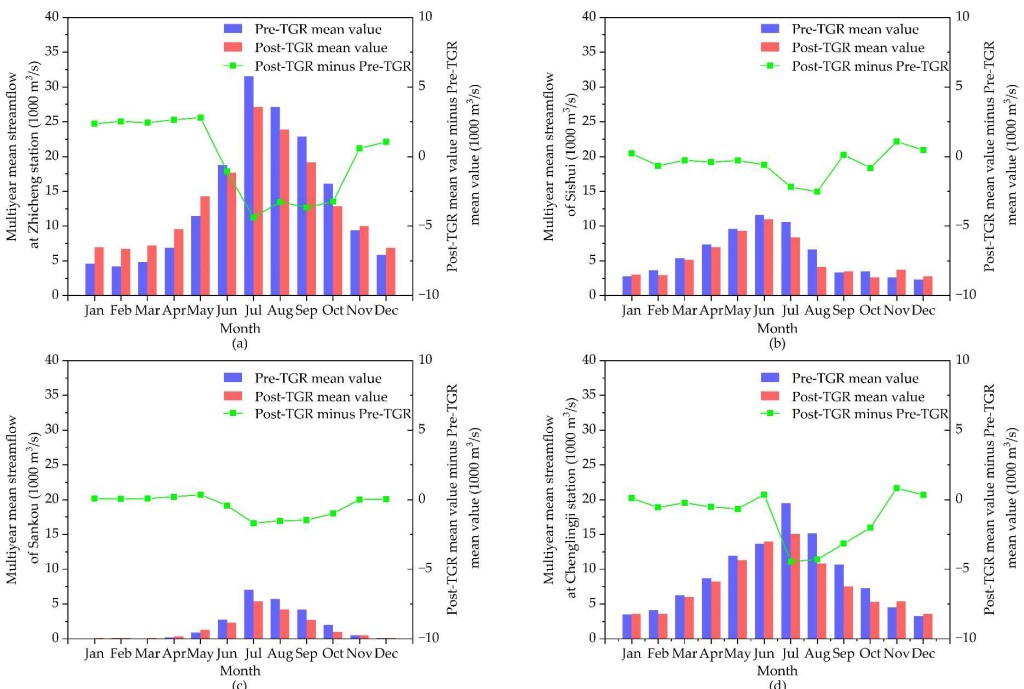

**Figure 9.** The observed multiyear mean streamflow: (**a**) Yangtze River; (**b**) Sishui; (**c**) Sankou; and (**d**) the outflow of Dongting Lake.

The relationship between the outflow and inflow of Dongting Lake is an important factor affecting the lake water level each month. Figure 10 displays a cumulative departure curve of the outflow and inflow of Dongting Lake in each month; the total inflow of Dongting Lake is the sum of the streamflows of Sishui and Sankou. The obvious change in the slope of the cumulative departure curve means that the relationship between the outflow and inflow in the pre-TGR period and the post-TGR period is inconsistent, and the water storage and drainage capacity of Dongting Lake has changed. The slopes of the cumulative departure curve in June, July, August, and October increased by 0.1, 0.04, 0.1, and 0.09, respectively, which shows that, under the condition of the same inflow, the drainage capacity of Dongting Lake increased in these months. The slopes of the cumulative departure curve in January, November, and December decreased by 0.12, 0.17, and 0.19, respectively, and this shows that, under the condition of the same inflow, the drainage capacity of Dongting Lake weakened in these months. The streamflow of the Yangtze River decreased from June to October and increased from November to May of the following year (Figure 9a). Changes in the streamflow of the Yangtze River will cause changes in the relative water level difference between Dongting Lake and the Yangtze River, resulting in

a change in the drainage capacity of Dongting Lake, which indicates that the change in the water level of Dongting Lake is partly caused by the change in the streamflow of the Yangtze River.

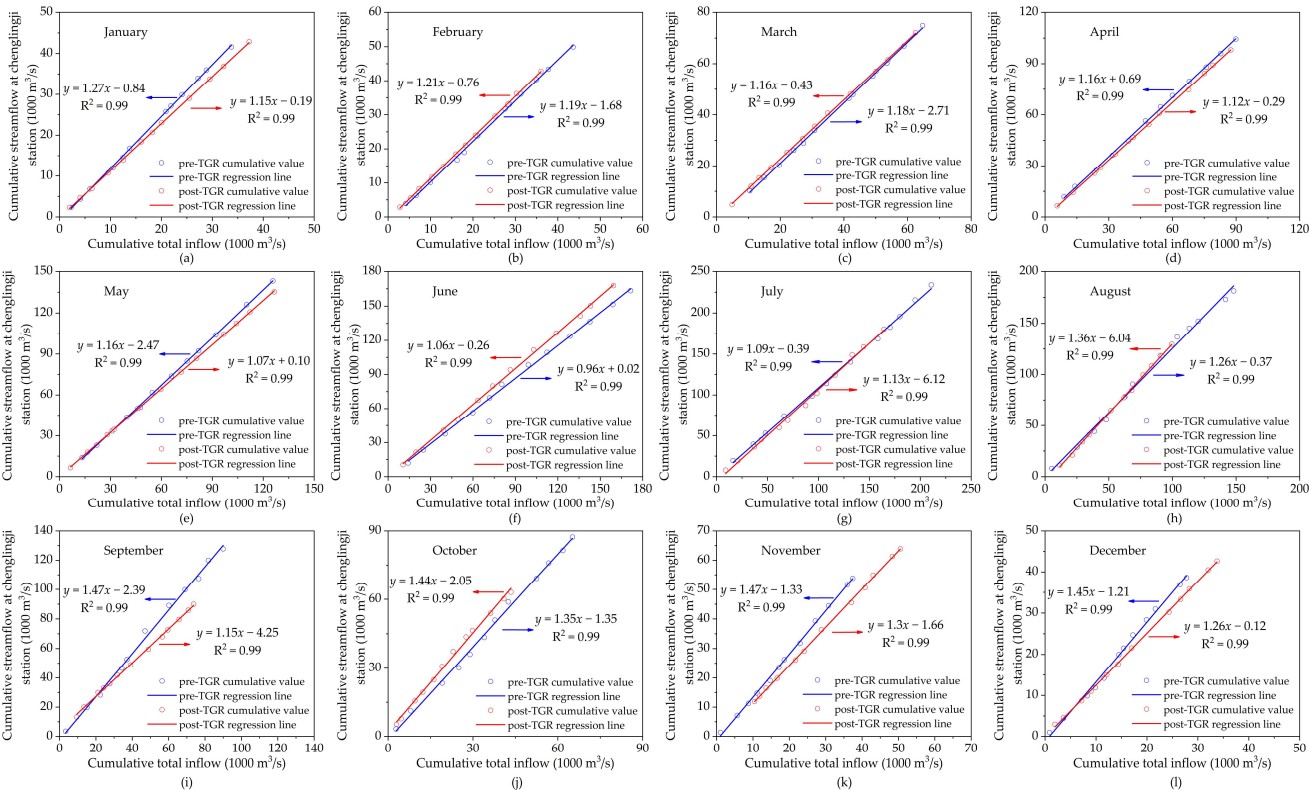

**Figure 10.** Cumulative departure curve of the outflow and inflow of Dongting Lake: (**a**) January; (**b**) February; (**c**) March; (**d**) April; (**e**) May; (**f**) June; (**g**) July; (**h**) August; (**i**) September; (**j**) October; (**k**) November; and (**l**) December.

### 4.4. Contribution of Changes in Streamflow and Topographic Conditions

Figure 11 displays the contribution of each driving factor to the water level changes in WDL, SDL, and EDL.

As for the changes in the streamflow of Sishui, they had a falling effect on the water levels of WDL, SDL, and EDL in July and August, and the falling effect was the strongest in August, with a falling effect of 0.69 m, 0.82 m, and 0.83 m, respectively. This made the water levels of WDL, SDL, and EDL increase slightly from November to January of the following year, with a rising effect of 0.19 m, 0.14 m, and 0.38 m, respectively. This is consistent with Figure 9b; the streamflow of Sishui most obviously decreased in July and August, by 2190 m$^3$/s and 2529 m$^3$/s, respectively, and there was a small increase in the streamflow from November to January of the following year.

As for the changes in the streamflow of the Yangtze River, they had a falling effect on the water levels of WDL, SDL, and EDL from July to October. For WDL, the falling effect was stronger in July and September, and it decreased by 0.43 m and 0.45 m, respectively. For SDL, the falling effect was stronger in July and September, and it decreased by 0.56 m and 0.64 m, respectively. For EDL, the falling effect was stronger in September and October, and it decreased by 0.94 m and 0.92 m, respectively. In the high-water stage, the strength of the falling effect was the strongest in EDL (0.67 m), followed by SDL (0.51 m), and the weakest in WDL (0.37 m). The changes in the streamflow of the Yangtze River had a rising effect on the water levels of WDL, SDL, and EDL from December to May of the following year, and the rising effect was 0.08 m, 0.04 m, and 0.43 m, respectively. This is mainly due to the water released by the TGR during this period, which led to the water level changes

in the downstream river and changes in the interaction between the Yangtze River and Dongting Lake. Since Sankou had close to no streamflow during this period (Figure 9c), the changes in the streamflow of the Yangtze River could only have a relatively obvious rising effect on the water level of EDL; this effect on WDL and SDL was almost negligible.

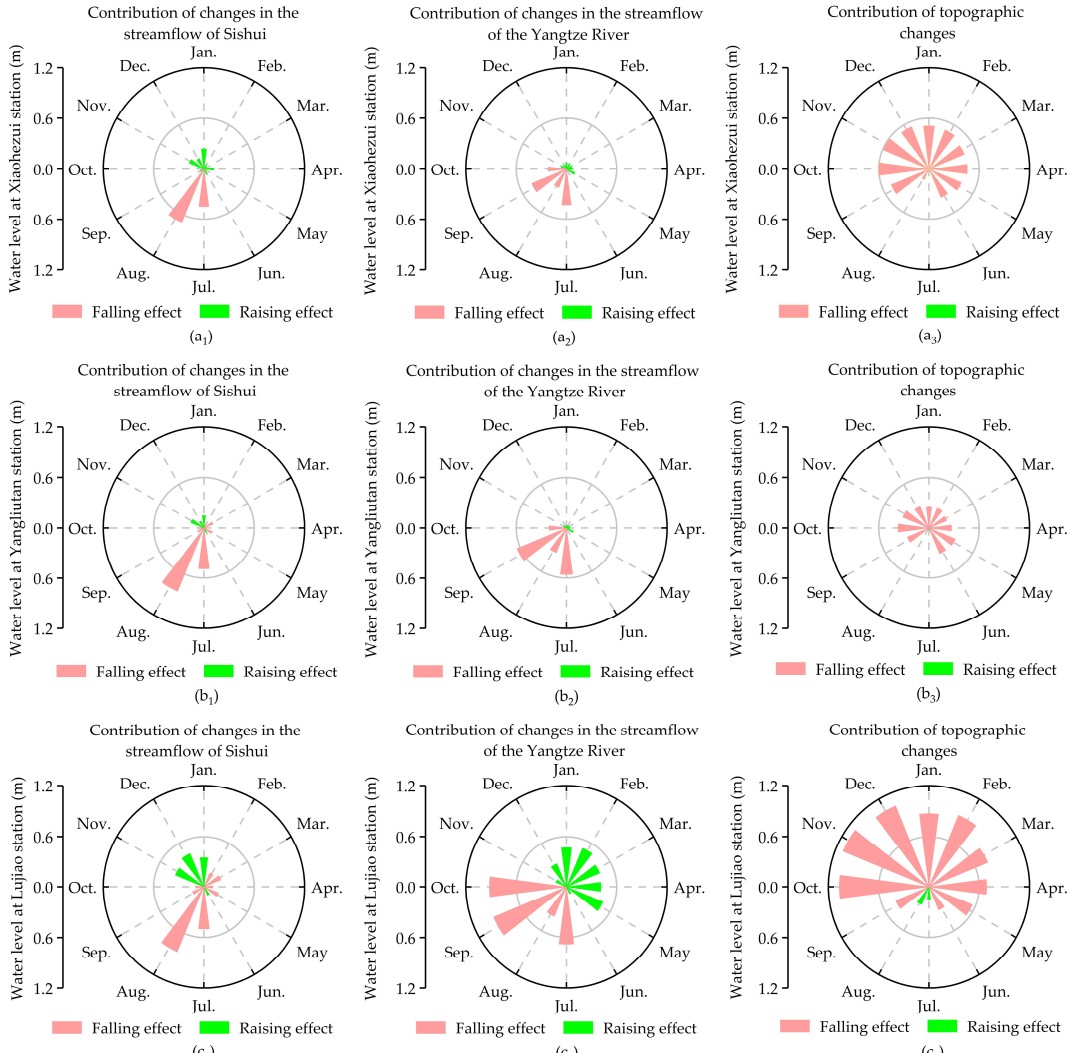

**Figure 11.** Contribution of each driving factor to water level changes in Dongting Lake: (**a₁**–**a₃**) Xiao-hezui station; (**b₁**–**b₃**) Yangliutan station; and (**c₁**–**c₃**) Lujiao station.

The topographic changes had a great falling effect on the water level of Dongting Lake during the water-falling stage, dry season, and water-rising stage, and their contribution to WDL was 0.6 m, 0.5 m, and 0.42 m, respectively; their contribution to SDL was 0.36 m, 0.26 m, and 0.31 m, respectively; and their contribution to EDL was 1.1 m, 0.9 m, and 0.52 m, respectively. The topographic changes mainly played a role in lowering the water level of Dongting Lake. From October to June of the following year, the strength of the falling effect was the strongest in EDL (0.81 m), followed by WDL (0.49 m), and the weakest in SDL (0.3 m).

## 5. Discussion

### 5.1. Topographic Changes in Dongting Lake

The operation of the TGR intercepted a large amount of sediment, resulting in a sharp decrease in the sediment content of the downstream river channel, and the downstream riverbed changed from a siltation state to an erosion state [28]. After 2003, the sediments

flowing into Dongting Lake from Sankou decreased significantly, and the lake area changed from a siltation state to an erosion state [2]. The topographic conditions of Dongting Lake changed dramatically [29]. In addition to the interception by the TGR, sand mining activities also cause changes in topographic conditions. Influenced by local policies, sand mining in the Yangtze River was controlled by management in 1998 [30]. As a result, many sand mining vessels entered Dongting Lake and Poyang Lake to carry out excessive and disorderly sand mining [31,32]. Since 2006, large-scale sand mining activities have occurred in Dongting Lake, and at the time, there were dozens of sand mining companies in the local area [33]. These sand mining activities can profoundly alter the conditions of the lakebed and the water storage capacity of Dongting Lake.

*5.2. Limitations and Future Research*

The changes in the water level of Dongting Lake are mainly affected by the inflow, the outflow, the shape of the lakebed, precipitation, evaporation, and water consumption by human activities. This study estimated the effects of changes in streamflow and topographic conditions. However, due to the limitations of the original data, the contribution of variables such as precipitation, evaporation, and water consumption by human activities to the water level changes in Dongting Lake was not analyzed separately. Wang et al. [34] found that after the operation of the TGR, the impact of precipitation and evaporation on the streamflow of Dongting Lake was more obvious, and the contribution rates of human activities, evaporation, and rainfall to the streamflow were 49.32%, 20.88%, and 29.80%, respectively. For the Dongting Lake Basin, the effect of changes in streamflow on the water level partially includes the influence of precipitation, evaporation, and human utilization. The complex relationship between these factors and the water level changes in Dongting Lake needs to be further clarified in subsequent studies.

**6. Conclusions**

This study analyzed the water level changes in Dongting Lake after the operation of the TGR and simulated the water level of Dongting Lake using the LSTMNN model. Based on scenario analysis, the contribution of changes in streamflow and topographic conditions to the water level changes in different areas of Dongting Lake was studied.

After the operation of the TGR, the different areas of Dongting Lake experienced various degrees of shrinkage, and the most obvious shrinkage occurred in EDL. With the TGR and reservoir upstream of Sishui intercepting floods, changes in the streamflow were found to be the main driving factors for the water level decline in WDL, SDL, and EDL in the high-water stage, and their contributions were 0.74 m, 0.97 m, and 1.16 m, respectively. Affected by sand mining activities and the change in erosion and deposition in the lake area, the topographic conditions of Dongting Lake changed obviously, and they had a great falling effect on the water level of Dongting Lake; the falling effect on the water levels from October to June of the following year was the strongest in EDL (0.81 m), followed by WDL (0.49 m), and the weakest in SDL (0.3 m).

The methods and results of this study can provide scientific reference for a cause analysis of Dongting Lake shrinkage, and they can provide useful information for the water resources planning and management of Dongting Lake.

**Author Contributions:** Conceptualization, F.H.; methodology, F.H. and J.Z.; software, F.H. and J.Z.; validation, F.H. and J.Z.; formal analysis, F.H., J.Z., X.S., S.H., Z.Q. and H.J.; investigation, F.H., J.Z., X.S., S.H., Z.Q. and H.J.; resources, F.H.; data curation, F.H. and J.Z.; writing—original draft preparation, F.H., J.Z. and F.Y.; writing—review and editing, F.H.; visualization, F.H., J.Z. and F.Y.; supervision, F.H.; project administration, F.H.; funding acquisition, F.H. All authors have read and agreed to the published version of the manuscript.

**Funding:** This research was funded by Water Conservancy Science and Technology Project of Hunan Province (XSKJ2021000-03; XSKJ2019081-05).

**Data Availability Statement:** Restrictions apply to the availability of these data. Data were obtained from Yangtze River Water Resources Commission of China.

**Conflicts of Interest:** The authors declare no conflict of interest.

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
