# Peer review of "Differentiating the Effects of Streamflow and Topographic Changes on the Water Level of Dongting Lake, China, Using the LSTM Network and Scenario Analysis"

_water, doi:10.3390/w15213742_

Round 1
Reviewer 1 Report
Comments and Suggestions for Authors
Dongting Lake is the second largest lake in China, and studying its water level changes has practical value for analyzing water resources. This study only investigated the effects of runoff and underwater topography on lake water level. The influencing factors on lake water level should consider factors such as precipitation, evaporation, groundwater inflow into the lake, and human utilization.
Comments on the Quality of English LanguageThe language in the manuscript needs improvement.
Reviewer 2 Report
Comments and Suggestions for Authors
It is not entirely clear from the text why the applied method does not take into account the influence also of changes/contributions of precipitation, evaporation, and water consumption by human activities on the water level change in Dongting Lake? Additional clarification would improve the text.
Reviewer 3 Report
Comments and Suggestions for Authors
The study on the water level changes in Dongting Lake is undoubtedly valuable, shedding light on the critical issue of declining water levels in this ecologically significant freshwater lake. However, there are several major revision comments that should be addressed to enhance the scientific rigor and clarity of the research.
Methodology and Model Explanation: The use of a long short-term memory neural network model is mentioned, but the paper lacks a thorough explanation of the model's architecture and how it was applied to simulate water levels. Providing more details on this aspect is crucial for readers to understand the validity and reliability of the model. Discuss the study “The effects of socioeconomic factors on particulate matter concentration in China's: New evidence from spatial econometric model” and MFFCG–Multi feature fusion for hyperspectral image classification using graph attention network.
Data Sources and Quality: The paper should include information on the sources of the data used, their temporal coverage, and data quality assessments. This is essential for replicability and for evaluating the robustness of the findings.
Discussion of Ecological Implications: Given that Dongting Lake is a critical habitat for migratory birds, it's important to discuss the ecological implications of the declining water levels. How might these changes affect the bird populations and the broader ecosystem? Incorporating this into the discussion section would add depth to the paper.
Statistical Significance and Confidence Intervals: When presenting the contributions of streamflow and topographic changes to water level decline, consider providing measures of statistical significance and confidence intervals to convey the robustness of the results.
Clarity and Structure: Ensure that the paper has a clear and logical structure, with well-defined sections, headings, and transitions between ideas. This will enhance the readability and flow of the manuscript.
Addressing these major revision comments will significantly strengthen the research and its contribution to our understanding of the factors affecting water levels in Dongting Lake and their implications for both the environment and water resource management.
Comments on the Quality of English LanguageThe study on the water level changes in Dongting Lake is undoubtedly valuable, shedding light on the critical issue of declining water levels in this ecologically significant freshwater lake. However, there are several major revision comments that should be addressed to enhance the scientific rigor and clarity of the research.
Methodology and Model Explanation: The use of a long short-term memory neural network model is mentioned, but the paper lacks a thorough explanation of the model's architecture and how it was applied to simulate water levels. Providing more details on this aspect is crucial for readers to understand the validity and reliability of the model. Discuss the study “The effects of socioeconomic factors on particulate matter concentration in China's: New evidence from spatial econometric model” and MFFCG–Multi feature fusion for hyperspectral image classification using graph attention network.
Data Sources and Quality: The paper should include information on the sources of the data used, their temporal coverage, and data quality assessments. This is essential for replicability and for evaluating the robustness of the findings.
Discussion of Ecological Implications: Given that Dongting Lake is a critical habitat for migratory birds, it's important to discuss the ecological implications of the declining water levels. How might these changes affect the bird populations and the broader ecosystem? Incorporating this into the discussion section would add depth to the paper.
Statistical Significance and Confidence Intervals: When presenting the contributions of streamflow and topographic changes to water level decline, consider providing measures of statistical significance and confidence intervals to convey the robustness of the results.
Clarity and Structure: Ensure that the paper has a clear and logical structure, with well-defined sections, headings, and transitions between ideas. This will enhance the readability and flow of the manuscript.
Addressing these major revision comments will significantly strengthen the research and its contribution to our understanding of the factors affecting water levels in Dongting Lake and their implications for both the environment and water resource management.
Round 2
Reviewer 1 Report
Comments and Suggestions for Authors
Reviewer 3 Report
Comments and Suggestions for Authors
accept
